# Barbed Pharyngoplasty for Snoring: Does It Meet the Expectations? A Systematic Review

**DOI:** 10.3390/healthcare11030435

**Published:** 2023-02-03

**Authors:** Antonio Moffa, Lucrezia Giorgi, Luca Carnuccio, Michele Cassano, Rodolfo Lugo, Peter Baptista, Manuele Casale

**Affiliations:** 1Integrated Therapies in Otolaryngology, Campus Bio-Medico University Hospital Foundation, 00128 Rome, Italy; 2School of Medicine, Campus Bio-Medico University, 00128 Rome, Italy; 3Unit of Measurements and Biomedical Instrumentation, Department of Engineering, Campus Bio-Medico University of Rome, 00128 Rome, Italy; 4Unit of Otolaryngology, University of Foggia, 71122 Foggia, Italy; 5Grupo Medico San Pedro, Department of Otorhinolaryngology, Monterrey 64660, Mexico; 6Department of Otorhinolaryngology, Clinica Universidad de Navarra, 31008 Pamplona, Spain

**Keywords:** snoring, obstructive sleep apnea, barbed sutures, barbed pharyngoplasty

## Abstract

To date, the use of barbed sutures for the surgical management of patients suffering from obstructive sleep apnea and snoring with retropalatal collapse and vibration has significantly increased. A systematic review was carried out, which included clinical studies that used barbed sutures for the treatment of snoring. A qualitative analysis, including six clinical studies, was conducted. Of these, five were studies on barbed pharyngoplasties, and one study involved a minimally invasive surgical procedure. The population consisted of 176 patients, aged 26 to 58 years old. Overall, the included studies showed a mean gain in the snoring Visual Analog Scale of 5.67 ± 1.88, with a mean preoperative value of 8.35 ± 1.17 and a postoperative value of 2.68 ± 1.27. No major complications were described. Given the lack and heterogeneity of this evidence, the conclusion calls for being cautious. In carefully selected snorers and obstructive sleep apnea patients, the use of barbed sutures could represent a valid therapeutic strategy for snoring, ensuring a statistically significant improvement in the subjective parameters. Further studies on a larger scale that assess the role of barbed pharyngoplasties in snoring surgery and more extended follow-up studies are needed in order to confirm these promising results.

## 1. Introduction

Snoring is a well-known common symptom in OSA patients, and its intensity is associated with this disease; however, it is not always accurate to assume that patients who snore have OSA. Indeed, an estimated 45% of adults snore only occasionally, whereas 25% snore regularly [1]. Typically, there is a big difference between genders: 40% of snorers are men, whereas 24% are women. Snoring is not only a “cosmetic” problem, but it can represent a health risk and can affect quality of life. Indeed, many snorers report non-refreshing sleep and a reduction in sleep time due to waking themselves up. Not getting enough sleep is linked to health problems, including obesity, diabetes, and heart disease. Habitual snorers are at greater risk of vascular disease [2,3]. The source of the sound of snoring is usually the presence of abnormalities of the soft palate or uvula. The passage of airflow leads to a vibration of the soft palate if it is too long or floppy, and this anomalous vibration creates the sound of snoring. Moreover, due to the resistance of the upper airway (UA) tract during sleep, snoring occurs during inspiration and rarely persists during expiration. The resistance increases as the UA narrows, producing the vibration and, consequently, the snoring sound [4]. During recent years, several surgical procedures performed in outpatient settings have been suggested to improve snoring, such as injection sclerotherapy, laser therapy, cautery procedures, radiofrequency ablation, and palatal implants. However, none of these have been shown to be effective for all snorers, as they mostly produce only temporary results. Most of these procedures act to increase palatal stiffening, creating scar tissue in the palate. Recently, Barbed Pharyngoplasty (BP) has been shown to reduce OSA and improve snoring [5]. This procedure involves the use of Barbed Sutures (BSs), which are threads that do not require knots and are capable of self-locking due to full-length directional projections that impart tensile strength within the tissues [6]. These threads are resorbable within 90–180 days, allowing tissue fibrosis to preserve the functional results. BSs can be used to suspend oral and pharyngeal tissue with the aim of improving the interaction between the thread and tissues [7]. Currently, in the literature, there are no reviews regarding the effects of BSs specifically for the treatment of snoring. In most studies, their effects on snoring were evaluated only secondarily compared to the effects on sleep apnea. The purposes of this review were first to provide a systematic overview of the current literature regarding the effectiveness of BS for the treatment of snoring, and then to describe the current findings regarding selection of the right patient, the possible side effects which can occur, an evaluation of the results, and how best to promote adherence.

## 2. Materials and Methods

### 2.1. General Study Design

The study was designed following the recommendations of the Centre for review and Dissemination’s Guidance for Under-taking Review in Health Care and is being reported in adherence with the Preferred Reporting Items for Systematic Review and Meta-Analyses (PRISMA) statement [8].

### 2.2. Data Source and Study Searching

An electronic search was performed on PubMed/MEDLINE, Google Scholar, and Ovid databases. An example of a search strategy is the one used for PubMed/MEDLINE: “Barbed” and “Snoring”; “Barbed” and “Pharyngoplasty”; “Barbed” and “Palatoplasty”; “Barbed” and “Anterior Pharyngoplasty”; “Barbed” and “Lateral Pharyngoplasty”; “Barbed” and “Expansion Sphincter Pharyngoplasty”; “Barbed” and “Suspension Pharyngoplasty”; “Barbed” and “Reposition Pharyngoplasty”, “Barbed” and “Snore Surgery”; “Barbed” and “Roman Blinds Technique”; “Barbed” and “Alianza Technique”; “Barbed” and “Habitual Snoring”; “Barbed” and “Heavy snorer”; “Barbed” and “Sleep Disturbance”; “Barbed” and “Palate Surgery”; “Barbed” and “Palatal stiffening”; “Barbed” and “Snoring Treatment”; “Barbed” and “Soft Palate”; and “Barbed” and “Soft Palatal Collapse”. All the searches were adjusted to fit the specific requirements for each database, with a cross-reference search in order to minimize the risk of missing relevant data. The last research was run in June 2021.

### 2.3. Inclusion/Exclusion Criteria

According to the PICOS acronym [9], we included the studies with the following characteristics: Patients (P), adults patients affected by snoring with and without OSA; Intervention (I), BP and in-office Elevoplasty procedures using BSs inserted into the soft palate; Comparison (C), pre- and post-treatment; Outcome (O), self-reported (e.g., snoring VAS) and the side effects; and Study design (S), both prospective and retrospective cohort studies. The exclusion criteria were: (1) studies not in English; (2) case reports, reviews, conference abstracts, letters, and pediatric studies; (3) studies with unclear and/or incomplete data; and (4) Studies evaluating only the effects of using BS threads for the treatment of OSA and not of nocturnal snoring. No publication date restriction was imposed.

### 2.4. Data Extraction and Data Analysis

All the articles were initially screened by title and abstract. Then, the full-text versions of each publication were assessed and those whose content was judged to not be strictly related to the subject of this review were excluded. Data extraction from the included studies was systematically performed using a structured form. A qualitative synthesis analysis was performed using the selected studies regarding the effects of different BP techniques.

### 2.5. Statistical Analysis and Summary of Findings

Due to the heterogenic reporting style and a lack of data in the studies included, it was not possible to conduct a statistical analysis or provide a quantitative summary of the findings. Thus, the effects on the individual outcomes and the overall quality assessments were solely narratively described. The authors of the included studies were not contacted for further information.

## 3. Results

The search criteria returned 35 articles, and 8 papers were removed as they were considered irrelevant or duplicates. As further screening occurred, 20 more papers were excluded, resulting in 7 articles that fulfilled the inclusion criteria. A flow diagram of the selection process is shown in Figure 1 (PRISMA flow diagram). The population in the included studies consisted of 202 patients aged 26 to 72 years old. The characteristics of these studies are reported in Table 1. Further descriptions of the studies can be found in Table 2. Of the included patients, only 52 were simple snorers, whereas the others suffered from mild to severe OSA. Overall, the included studies showed a mean improvement in the snoring Visual Analog Scale (VAS) of 5.52 ± 1.91, with a mean preoperative value of 8.28 ± 1.18 and a postoperative value of 2.75 ± 1.29.

### 3.1. Barbed Pharyngoplasties

Salamanca et al. [15] performed Barbed Anterior Pharyngoplasty (BAP) on 24 patients with anterior–posterior palatal collapse: 17 with simple snoring and 7 with mild OSA. At the end of the treatment, the authors observed a statistically significant reduction in the snoring VAS for all the treated patients. For the same parameter, a significant improvement was also reported by Elbassiouny et al. [14] (9.4 ± 1.6 vs. 1.7 ± 3.2), who performed a modified barbed soft palatal posterior pillar webbing flap palatopharyngoplasty on 21 severe OSA patients with loud snoring (71% presented with anteroposterior collapse, whereas 29% presented with concentric collapse).

Later, Mantovani et al. [13] described their preliminary experience with the Alianza technique in 19 mild-to-moderate OSA patients with concentric palatal collapse. After six months, the authors showed a substantial reduction in snoring VAS (9.5 ± 0.7 vs. 2.1 ± 1.7). Using the same technique, Casale et al. [10] recently showed a statistically significant decrease in the mean post-operative snoring VAS after at least 6 months of follow-up (7.85 ± 1.23 vs. 3.2 ± 1.7).

Babademez et al. [12] performed Barbed Reposition Pharyngoplasty (BRP) on 17 OSA patients and modified BRP (MBRP) on another 17 patients. The authors observed a significant decrease in the snoring VAS both using BRP and the modified BRP. In particular, the reduction in snoring VAS was greater in the MBRP group than the BRP group without any significant differences.

Lastly, an improvement in the snoring VAS was found by Carrasco et al. [11], who performed an MBRP on 26 mild-to-severe OSA patients with laterolateral collapse.

### 3.2. Minimally Invasive Surgical Procedure

Friedman et al. [16] performed palatal foreshortening and stiffening in order to reduce the snoring severity in patients with chronic disruptive snoring. Each subject was treated with an in-office elevoplasty procedure using three fully resorbable (polydioxanone) BS threads inserted into the soft palate under local anesthesia. They showed a significant decrease in the snoring VAS at 30, 90, and 180 days after the procedure.

### 3.3. Side Effects

Only minor complications were recorded in the studies included in this review, and none were recorded in the minimally invasive surgical procedure study. The observed complications were partial knot extrusions (30.11% of the patients), temporary velopharyngeal insufficiency (1.7%), excessive postnasal discharge due to shortening of the uvula (2.27%), mucosal granulomas (0.57%), anterior pharyngoplasty dehiscence (3.41%), post-tonsillectomy hemorrhaging (1.14%), mild oropharyngeal pain, and swallowing difficulty [10,11,12,14,15]. For these last two, the number of patients were not specified. All the complications were described as “temporary”. These results are reported in Table 3.

## 4. Discussion

Snoring is a highly prevalent condition, afflicting 40 to 60% of the adult population, and affects the quality of sleep of both patients and their bed partners. However, it is less known that it may also cause daytime consequences, such as sleepiness, neurocognitive impairment, and mood disturbances. Because it is also a precursor of OSA, it could predispose to cardio- and cerebro-vascular diseases and may lead to car accidents [17].

It is known that the noise of snoring mainly originates at the soft palate level and surrounding structures due to a combination of anatomical and neuromuscular factors causing a critical sleep-related reduction in upper airway airflow [18].

Weight loss and other lifestyle modifications, such as the avoidance of alcohol and sedatives, are considered a conservative approach to managing snoring. In addition to these, pharmacological or surgical solutions should be considered in order to resolve any associated nasal obstruction [19]. However, only 12–34% of patients benefit from nasal surgery alone, so the effectiveness of this treatment is controversial [20,21].

In recent years, many resective surgical procedures have been proposed, including uvulopalatopharyngoplasty [22] and laser-assisted or radiofrequency-assisted uvulopalatoplasty [23], all of which involve severe postoperative discomfort and morbidity. On the other hand, non-resective techniques, such as radiofrequency soft palate volume reduction, injection scleroplasty, or expensive palatal implants (Pillar^®,^, Medtronic, Minneapolis, United States of America), have some drawbacks as they are only transient and require revision [24].

In order to reduce the invasiveness of surgical procedures, which aim to increase upper airway tension, and because permanent threads in the soft palate are well tolerated, Mantovani et al. [7] introduced BSs, which are special knot-free tissue closure devices that allow the homogeneous distribution of tensile closure forces.

The use of BSs for the treatment of retropalatal collapse and vibration in patients suffering from snoring and OSA has increased significantly in recent years. Many surgeons have discovered the advantages and properties of BSs, which has facilitated the revision and improvement of popular surgical pharyngoplasty techniques [5].

BSs are not frequently used to treat simple snorers. Conversely, they are frequently used for OSA patients. From this focused review, it is clear that BSs, regardless of the technique used, are safe and effective in reducing not only OSA but also snoring. In particular, all the studies evaluated in this review showed a statistically significant improvement in the snoring VAS post-surgery. However, the studies were very heterogeneous, as they included both simple snorers and patients with mild-to-moderate OSA. Ideally, the effects of BSs should be verified only on patients with simple snoring, but this type of study is very rare. All the authors performed BP alone, whereas Babadamez et al. [12] performed it as a part of multilevel surgery. Therefore, it is more difficult to assess the role and power of BP on palatal snoring in multilevel surgery. In addition, the follow-up period ranged from 1 to 9 months. The world of BPs is very heterogeneous. There are many different surgical techniques, each characterized by their own peculiarities, as was shown recently in a systematic review [5] that investigated the use of BSs. This review stemmed from the need to comprehensively assess the effects of BSs on snoring, regardless of the technique used. Salamanca et al. [15] showed the effects of Barbed Anterior Pharyngoplasty, characterized by the passage of the Barbed Sutures only at the level of the soft palate without any work on the lateral walls of the pharynx. Mantovani [13] used the Alianza technique, which is composed of barbed anterior and lateral pharyngoplasty, without any cutting or weakening of the palatopharyngeal muscle. Both of these studies included patients with no tonsils or with very small tonsils. Carrasco Llatas et al. [11] and Babademez et al. [12] analyzed the effects of BRP and Modified BRP, which provide stable retraction of the pharyngeal soft tissue through traction, while preserving the mucosal and muscle tissue, and a stable suspension of the palatopharyngeal muscle in a lateral and anterior position to the pterygomandibular raphe, expanding the lateral walls of the oropharynx. In order to carry out this technique, it Is necessary to perform tonsillectomy if the tonsils are present. Friedman [16] proposed an in-office Elevoplasty procedure, whereby three fully resorbable (polydioxanone) barbed suture implants are inserted into the soft palate under topical and local infiltration anesthesia. Finally, Elbassiouny [14] used modified barbed soft palatal posterior pillar webbing flap palatopharyngoplasty, in which the soft palatal redundancy was managed as a separate palatoplasty and lateral pharyngeal wall tension was achieved by fashioning two flaps.

Moreover, it should be kept in mind that assessing the severity of snoring and how annoying it is to a patient’s bed partner is not easy, as snorers are unable to provide a self-assessment and statements from bed partners may not be reliable [25].

An objective snoring measurement is not frequently used in common clinical practice. Solid parameters are lacking and there are only few possible objective measures that can performed, such as the snoring index (number of snoring sounds per hour), the percentage of sleep time or the total snoring time [26,27], and psychoacoustic parameters, such as annoyance [28]. However, none of the studies included in this review used objective parameters. Instead, they evaluated snoring changes using the VAS (from 1 to 10), except in one study where the snoring scale (from 1 to 5) was adopted [11]. This aspect represents a very strong limitation of these studies. Therefore, it is necessary that future studies evaluating the effects of BP on snoring take into consideration not only subjective parameters, but also objective parameters.

The surgical treatment of snoring consists of several procedures that aim to shorten and/or reduce the soft palate, or that stiffen the soft palate itself to decrease vibrancy. BSs allow more consistent tension control using knotless continuous closure technology. Finally, in this review, we report the only study in the literature that describes an outpatient surgical technique with BSs [16].

Regarding patient selection, Friedman et al. [16] did not perform DISE, unlike the other authors. However, this study is the only one conducted on individuals with simple snorers. The other authors did not specify the DISE classification system used, except for Mantovani et al. [13] (NOHL [29]) and Babademez et al. [12] (VOTE [30]). In addition, Elbassiouny et al. [14], Mantovani et al. [13], and Babademez et al. [12], also performed the Müller maneuver.

It is essential to characterize, in detail, patients with snoring through DISE in order to understand the site and pattern of obstruction and/or collapse and vibration.

No major complications were recorded in this review. Nevertheless, only a few minor complications were described, such as partial knot extrusions [10,11,13,15], temporary velopharyngeal insufficiency [11,14], excessive postnasal discharge due to shortening of the uvula [14], mucosal granulomas [13], anterior pharyngoplasty dehiscence [10,13], mild oropharyngeal pain [12], swallowing difficulty [12], and post-tonsillectomy hemorrhaging [11]. Both adequate post-operative pain management and a minimally invasive approach play key roles. Rinaldi et al. [31] showed that devices operating at low temperatures (such as plasma surgery systems) could reduce thermal damage to the tissues, thus facilitating wound recovery, especially for barbed anterior pharyngoplasty.

None of the treatments solved snoring in all the patients. As a first step, the medical history of the patient and careful physical examination can help to determine which therapy may be helpful. However, the idea of a “step therapy” with incremental improvements should be kept in mind in these types of patients. Indeed, multiple interventions may be needed in order to achieve success and a peaceful coexistence for bed partners [4].

Patients suffering from snoring would like to solve the problem immediately, but it is necessary to clarify that this is not possible, and that treating snoring is not easy.

BP showed many advantages over traditional surgery, as it is a bloodless, safe, quick, and repeatable technique. Moreover, stiffening of the soft palate is achieved through muscular folding and suspension to create a rigid hold, while maintaining rigorous anatomic respect for the palatal fibromuscular structures.

Lastly, the use of BSs ensures a uniform distribution of tissue tension over the entire length of the thread, providing excellent tissue grip and avoiding the need for tying knots without altering any muscle function. In addition, the action of the BSs is replaced over time by solid and stable tissue healing.

However, currently in the literature, there are only short-term studies on the efficacy of BP (monolevel and multilevel surgery) and no long-term studies evaluating the effects and complications of this technique have been performed [5]. Due to the data analyzed in this review, in the future, it could be very interesting to use barbed sutures to treat simple snorers by preferentially performing the surgical procedure under local anesthesia in order to shorten and stiffen the soft palate.

Thus, many BPs were intended to dampen the vibrations from the soft palate (either including or not including the uvula) and the lateral pharyngeal wall. Usually, the chances of snoring are mainly caused by the length and the floppiness of the soft palate (including the uvula) and the speed and turbulence of the retropalatal airflow through a narrow space. From an anatomical and functional point of view, the following steps are necessary in order to reduce snoring:Shortening of the soft palate (the distance between posterior spine and the free edge), which is minimal and usually additional steps, such as midline crossing sutures, are recommended;Dampening of the wave of vibration by reducing the elongated uvula;Stiffening of the thin palate, which tends to be less rigid and more prone to producing vibrations, by pulling out the insertions around the palate;Increasing the posteriorly collapsed palate (flat pharynx) or the funnel-shaped retropalatal area [32].

Compared with traditional pharyngoplasty techniques, such as uvulopalatopharyngoplasty and laser/radiofrequency-assisted uvulopalatoplasty, BPs, particularly BRP and Alianza, are conservative and minimally invasive techniques that selectively act on the palatopharyngeal mucosa, fibrous tissues, and muscles. These techniques can be combined with other surgical procedures, such as septoplasty and turbinoplasty, when additional anatomical abnormalities that compromise physiological airflow are present. In addition, BPs are cost-effective and well tolerated by patients who report only moderate oropharyngeal pain, which disappears quickly and spontaneously. No other significant postoperative morbidities or complications were noted.

This technique also appears to provide immediate relief from snoring and, theoretically, could be performed in selected patients under local anesthesia in a one-day surgery setting after proper training and with the use of devices that reduce surgical time.

It is mandatory to consider the surgical management of snoring as part of a multimodal approach in order to achieve successful treatment for snorers and OSA patients. The success reached through BPs could ameliorate the use of Oral Myofunctional Therapy (OMFT). In fact, it can be applied to both treated (e.g., soft palate) and untreated (e.g., facial and tongue muscles) muscle segments. In either case, OMFT should not be administered before surgical wound restoration, which is usually achieved within 1 month. OMFT induces upper airway remodeling, by reducing the parapharyngeal fat pads and decreasing the amount of fat in tongue muscle fibers, and improves airway patency during sleep. Five clinical studies, two meta-analyses, and one systematic review have investigated the role of OMFT in reducing snoring. The available evidence shows the positive effect of OMFT in reducing snoring, as measured by both objective (PSG) and subjective (scale) assessments [33]. It is possible to perform OMFT exercises through three different modalities: conventional outpatient sessions, outpatient sessions with electrical stimulation (eXCiteOSA) [34], and telemedicine with apps (AirwayGym App, Apnea Bye Sociedad Limitada, Sevilla, Spain) [35].

## 5. Conclusions

Given the lack and heterogeneity of this evidence, and the limited number of studies investigating small populations with short follow-up periods, the results of this review must be considered with caution. Therefore, in carefully selected snorers and OSA patients, BP could represent a valid therapeutic strategy for snoring, ensuring significant improvements in the subjective parameters. Further studies on a larger scale that assess the role of BP in snoring surgery, as well as more extensive follow-up studies, are needed in order to confirm these promising results.

## Figures and Tables

**Figure 1 healthcare-11-00435-f001:**
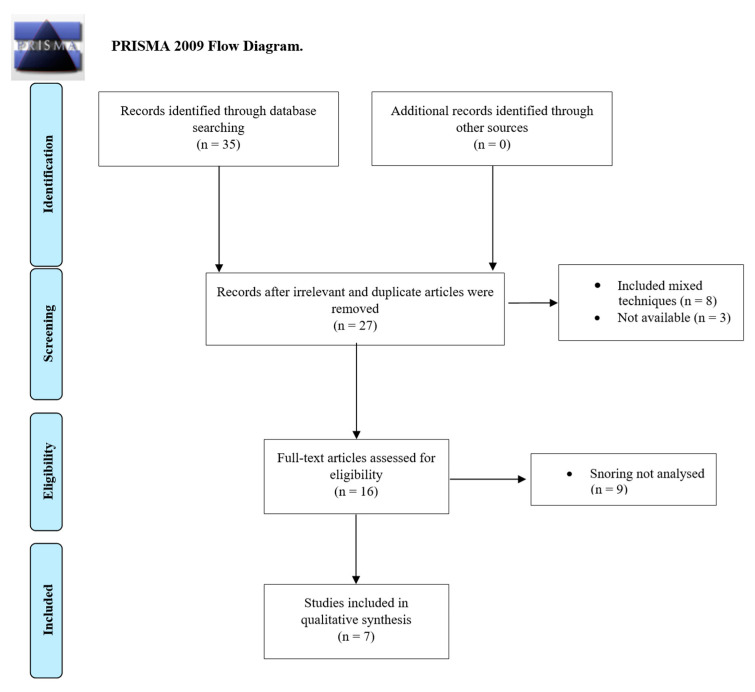
Flowchart of the paper selection process (based on PRISMA guidelines).

**Table 1 healthcare-11-00435-t001:** General characteristics of the included studies.

**Barbed Pharyngoplasties**									
**Author (Year)**	**Country**	**Study Design**	**DISE Findings**	**Snoring/OSA**	**Num. Patients**	**Mean Age (Years)**	**Sex (M:F)**	**BMI**	**Follow-Up (Month)**
Casale 2022 [10]	Italy	Prospective	Circular palatal collapse	Moderate to severe OSA	26	52.7 ± 9.2	26:0	28.1 ± 3.3	6.5 ± 2
Carrasco Llatas 2020 [11]	Spain	Retrospective	Retropalatal collapse/collapse of the pharyngeal lateral walls	Mild OSA (*n* = 5) Moderate OSA (*n* = 10) Severe OSA (*n* = 11)	26	42.5 ± 11.5	20:6	29.1 ± 4.3	3–6
Babademez 2019 [12]	Turkey	Prospective	-	Mild to moderate OSA	34	BRP = 39.4 ± 7.5 (26–58) MBRP = 40.6 ± 2.4 (27–58)	BRP = 14:3 MBRP = 15:2	BRP = 27.9 ± 3 MBRP = 27 ± 2.5	8 (6–9)
Mantovani 2017 [13]	Italy	Longitudinal	Concentric collapse	Mild to moderate OSA	19	43.8 ± 8.8	18:01	26.2 ± 2.7	6
Elbassiouny 2016 [14]	Egypt Kuwait	Prospective Single-centerun controlled case series	71% anteroposterior collapse, 29% concentric collapse	-	21	31.3 ± 4.8 (27–46)	3.2:1	28.3 ± 4.7	6
Salamanca 2014 [15]	Italy	Not specified	Anteroposterior palatal obstruction and vibration	Mild OSA (*n* = 7) Heavy snoring (*n* = 17)	24	46 (34–55)	2.4:1	28.6 (24.3–30.1)	1
**Minimally Invasive Surgery**									
**Author (Year)**	**Country**	**Study Design**	**DISE Findings**	**Snoring/OSAS**	**Num. Patients**	**Mean Age (Years)**	**Sex (M:F)**	**BMI**	**Follow-Up (Month)**
Friedman 2019 [16]	USA	Prospective Multicenter	-	Simple snoring	52	-	33:19		2

**Table 2 healthcare-11-00435-t002:** Summary of studies specifically involving the use of BS for the treatment of snoring.

**Barbed Pharyngoplasties**					
**Author (Year)**	**Type of Surgery**	**Surgical Technique**	**PRE Snoring VAS**	**POST Snoring VAS**	** *p* ** **-Value**
Casale 2022 [10]	Monolevel	Alianza	7.85 ± 1.23	3.20 ± 1.70	<0.01
Carrasco Llatas 2020 [11]	Monolevel	MBRP	3.3 ± 0.9 *	1.9 ± 1.3 *	0.004
Babademez 2019 [12]	Multilevel	BRP (*n* = 17) MBRP (*n* = 17)	BRP: 6.2 ± 1.9 MBRP: 8 ± 1.5	BRP: 2.2 ± 1 MBRP: 1.8 ± 0.8	BRP: < 0.0001 MBRP: < 0.0001
Mantovani 2017 [13]	Monolevel	Intraoperatively modulated technique, then with the Roman blind technique, barbed anterior pharyngoplasty for antero-poster collapse, or both (the Alianza technique)	9.5 ± 0.7	2.1 ± 1.7	<0.001
Elbassiouny 2016 [14]	Monolevel	Modified barbed soft palatal posterior pillar webbing flap palatopharyngoplasty	9.4 ± 1.6	1.7 ± 3.2	<0.005
Salamanca 2014 [15]	Monolevel	BAP	9.2	2.9	-
**Minimally Invasive Surgery**					
**Author (Year)**	**Type of Surgery**	**Surgical Technique**	**PRE Snoring VAS**	**POST Snoring VAS**	** *p* ** **-Value**
Friedman 2019 [16]	Snoring	Elevoplasty	7.81 ± 1.59	5.40 ± 2.28	<0.001

* The authors considered a different scale for the snoring VAS, from one to five.

**Table 3 healthcare-11-00435-t003:** Complications.

Complication	Num. Patients	% of Patients
Knot extrusion	53	30.11
Temporal velopharyngeal insufficiency	3	1.7
Anterior pharyngoplasty dehiscence	6	3.41
Post-tonsillectomy hemorrhage	2	1.14
Excessive postnasal discharge	4	2.27
Mucosal granulomas	1	0.57
Mild oropharyngeal pain	Not specified	-
Swallowing difficulty	Not specified	-
Total	69	39.20

## Data Availability

The datasets generated during and/or analyzed during the current study are available from the corresponding author on reasonable request.

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
