# Peer review of "Barbed Pharyngoplasty for Snoring: Does It Meet the Expectations? A Systematic Review"

_healthcare, 2023, doi:10.3390/healthcare11030435_

Round 1

Reviewer 1 Report

BP is a technique that can be performed in several ways. 1) With or without tonsillectomy. 2) With or without going through the palatopharyngeus muscle. It is not clear which techniques were used in the different studies. The authors must provide more clarity on this, otherwise it will not be clear what the role of the barbed suture itself is.

Author Response

Reviewer 1

BP is a technique that can be performed in several ways. 1) With or without tonsillectomy. 2) With or without going through the palatopharyngeus muscle. It is not clear which techniques were used in the different studies. The authors must provide more clarity on this, otherwise it will not be clear what the role of the barbed suture itself is.

Answer 1:

Thank you very much for your comment. In the table 2 we specified all the surgical techniques used for each article involving the use of BS for the snoring treatment. However, as you suggested, we clarified this aspect. Therefore we added in the discussion section the following sentences:

“Salamanca et al [10] showed the effects of Barbed Anterior Pharyngoplasty characterized by the passage of the Barbed Sutures only at the level of the soft palate without any work at the lateral walls of the pharynx. Mantovani [12] used the Alianza technique which is composed of barbed anterior and lateral pharyngoplasty, without any cutting or weakening of the palatopharyngeal muscle. Both studies had patients with no tonsils or with very small tonsils Carrasco Llatas et al [14] and Babademez et al [13] analyzed the effects of BRP and Modified BRP, which provides the stable retraction of the pharyngeal soft tissue due to the traction with preserving the mucosal and muscle tissue and a stable suspension of the palatopharyngeal muscle in a lateral and anterior position to the pterygomandibular raphe for expansion of the lateral walls of the oropharynx. In order to carry out this technique it is necessary to perform a tonsilectomy if the tonsils are present. Friedman [15] proposed in-office Elevoplasty procedure whereby three, fully resorbable (polydioxanone), barbed suture implants inserted into the soft palate under topical and local infiltration anesthesia. Finally, Elbassiouny [11] used Modified barbed soft palatal posterior pillar webbing flap palatopharyngoplasty, the soft palatal redundancy was managed as a separate palatoplasty while creating lateral pharyngeal wall tension by fashioning two flaps.”

Reviewer 2 Report

This is a narrative review regarding the effectiveness of Barbed Pharyngoplasty (BP)involving the use of Barbed Suture (BS), for the treatment of snoring. The authors selected 6 articles from 34 of the literature; they found a total 176 patients who underwent BP, out ofwhom 52 were simple snorers.

The authors characterized their study as a systematic review, whereas it actually presents a narrative review of the literature and does not follow systematic review methodology. Specifically, there is no valid comparison with another treatment, but statistical measures refer to pre- and post-BP. The population is also significantly heterogeneous, including both patients with OSA and simple snorers without separate analysis.. In the Discussion Section, the authors recognized that in these few articles there was heterogeneity regarding the different surgical techniques used and the absence of objective parameters for the validation of snoring, which are significant limitations.

The aim of this review was to introduce the BP procedure as an alternative therapeutic method to reduce OSA and improve snoring. The authors described the VAS score before and after the procedure, the follow-up period and the (rare) complications. However, this study would be more appropriately presented as a Letter to the Editor, instead of review type, as it does not appear to significantly add to the current literature with clinical nor research implications.

Author Response

Reviewer 2

This is a narrative review regarding the effectiveness of Barbed Pharyngoplasty (BP)involving the use of Barbed Suture (BS), for the treatment of snoring. The authors selected 6 articles from 34 of the literature; they found a total 176 patients who underwent BP, out of whom 52 were simple snorers.

The authors characterized their study as a systematic review, whereas it actually presents a narrative review of the literature and does not follow systematic review methodology. Specifically, there is no valid comparison with another treatment, but statistical measures refer to pre- and post-BP. The population is also significantly heterogeneous, including both patients with OSA and simple snorers without separate analysis.. In the Discussion Section, the authors recognized that in these few articles there was heterogeneity regarding the different surgical techniques used and the absence of objective parameters for the validation of snoring, which are significant limitations.

The aim of this review was to introduce the BP procedure as an alternative therapeutic method to reduce OSA and improve snoring. The authors described the VAS score before and after the procedure, the follow-up period and the (rare) complications. However, this study would be more appropriately presented as a Letter to the Editor, instead of review type, as it does not appear to significantly add to the current literature with clinical nor research implications.

Answer 2:

I have carefully read your comment. I am aware of your concerns about the heterogeneity of the studies included and the small number. However, they are the only studies in the literature dealing with the topic.

To date, in the literature Barbed Sutures have always been used to treat OSA and not to treat snores. The effect of Barbed Sutures on simple snoring has always been valued secondarily. The aim of our work was to underline the importance of using Barbed Sutures also to improve snoring as well as sleep apnea. Regarding the type of our manuscript, we followed our usual strict rules including all the articles which evaluated the effects of Barbed Sutures on the snoring. This article may be important because it paves the way for new treatments for simple snoring. As you are well aware, treating snoring is not simple and represents a challenge. The use of Barbed Sutures under general or local anesthesia could be very useful as supported by these promising results.